# Comment on Ceruti et al. Temporal Changes in the Oxyhemoglobin Dissociation Curve of Critically Ill COVID-19 Patients. *J. Clin. Med.* 2022, *11*, 788

**DOI:** 10.3390/jcm11123376

**Published:** 2022-06-13

**Authors:** Dieter Böning, Wilhelm Bloch, Wolfgang M. Kuebler

**Affiliations:** 1Institute of Physiology, Campus Mitte, Charité Medical University of Berlin, 10117 Berlin, Germany; wolfgang.kuebler@charite.de; 2Department of Molecular and Cellular Sport Medicine, Institute of Cardiovascular Research and Sport Medicine, German Sport University Cologne, 50933 Cologne, Germany; w.bloch@dshs-koeln.de

**Keywords:** hemoglobin oxygen affinity, half saturation pressure, P_50_, 2,3-Biphosphoglycerate, Hill equation

## Abstract

Ceruti et al. describe in their article very low standard half saturation pressures (P_50_) in COVID-19 patients, calculated with the Dash et al. equations. By using the Hill equation and Severinghaus‘ coefficients we obtained normal values. The authors who do not present a pathophysiological cause for their results should explain this discrepancy. Independent of the absolute values a continuous moderate decrease of P_50_ in the surviving patients might be of clinical importance.

Very recently Ceruti et al., published an interesting article about “Temporal Changes in the Oxyhemoglobin Dissociation Curve of Critically Ill COVID-19 Patients“ [1]. Similar to Vogel et al. in 2020 [2], they found a low standard half saturation pressure P_50_ in patients undergoing invasive mechanical ventilation which decreased more during the subsequent days in patients with a good prognosis. Surprisingly, however, their P_50_ values calculated with the Dash et al. equations [3] are unexpectedly low (around 20 mmHg, single values as low as 11.1 mmHg). Ceruti et al., present no convincing explanation as well as no comparison with healthy subjects. In many years of investigations, we have never seen such low values in human blood of adults with normal Hb A. Standard P_50_ for human red cells completely depleted of 2,3-Biphosphoglycerate (2,3-BPG) decreases to only 16 mmHg [4].

In Figure S2 (Supplementary Materials) [1], the authors compare the P_50_ calculated with the Hill equation according to Severinghaus [5] and with the Dash equations [3]. While the difference is small (maximally 2mm Hg) or even negligible above 16% saturation, the Dash values dramatically decrease at lower saturations. We cannot see any advantage of the Dash equations. Furthermore, Severinghaus‘ equations have been in use for decades, for comparison with previous investigations they are much more suitable.

This caused us to recalculate the evaluation using the Hill equation [6]
log SO_2_/(100 − SO_2_) = n × PO_2_ + K

According to general experience (e.g., [7]), the Hill coefficient n is rather constant during registration of an ODC in native blood (approximately 2.7). Therefore, it is possible to calculate a standard P_50_ from one pair of PO_2_ corrected to standard conditions (pH 7.4, Base excess 0 mequ/L, 37 °C) by the factors published by Severinghaus [5] and SO_2_. We applied this procedure to Ceruti‘s data in Table 1 for all patients (probably the means of initial values; unfortunately, the time of sampling is not communicated) and obtained an average P_50_ of 26.9 mmHg, very similar to Severinghaus‘ mean standard value 26.7 mmHg. For an assumed n = 3.0, P_50_ rises to 30.0, for 2.5, it decreases to 24.6 mmHg. We also evaluated the individual values for each patient in Table 2 and obtained 26.7 ± 1.9 SD mmHg. These values, ranging from 22.9 to 32.3 mmHg, are much higher than the published ones in Ceruti’s paper. Only 7 single values are lower than 25 mmHg. We also evaluated the data for the 25th and 75th percentile of SO_2_ in Table S1. These P_50_ are lower (23.0 and 22.5 mmHg) but still higher than the means in Ceruti‘s paper (20.6 at the beginning of the stay in the intense care unit, 18.7 mmHg at the end). Can Ceruti et al. explain the differences between their published values and our calculations?

The calculation of P_50_ from arterial samples is not recommended because of the flattening of the curve which markedly increases the scattering of PO_2_ [8]. We have, therefore, additionally excluded samples with SO_2_ higher than 95%, but this did not change the mean standard P_50_.

While the P_50_ values calculated by Ceruti et al. are obviously too low, the temporal decrease in their surviving patients may exist. The probably systematic error should be similar for all calculations during the stay in the intense care unit. Obviously, however, a control group of patients with other illnesses like in the paper of Vogel et al. [2] would have clarified the findings.

Also unanswered is the question: What causes the left shift of the curve during the illness? Vogel et al. [2,9], too, have published low P_50_ in heavily ill COVID-19 patients by applying Hill`s equation and the usual correction according to Severinghaus [5] to their measurements. Others detected no significant changes, but a tendency for a left shift or a large unexplained scattering is visible (reviewed and discussed by us [10,11,12,13]). The causes are not known. We suggested, originally, a high concentration of Met-Hb, but this was not increased in Vogel’s as well as in Ceruti‘s patients. Perhaps the amount of 2,3-Biphosphoglycerate in the red cells decreases in COVID-19 patients, but there exists only one paper with measurements in not heavily ill patients [14]. Additionally, the results are not clear (only arbitrary units are communicated, these are higher than in the control patients). Both groups were slightly anemic which usually causes an increase in [2,3-BPG] with normal acid–base-status (reviewed in [11]). Finally, drugs used for treatment might cause P_50_ changes. Recently, Woyke et al. [15] demonstrated in in vitro experiments that nebulized epoprostenol (a potent vasodilatory prostaglandin) significantly decreased P_50_ by approximately 3 mmHg thereby increasing Hb-O_2_ affinity. As inhaled epoprostenol is also used for the treatment of COVID-19 (ClinicalTrials.gov Identifier: NCT04452669, accessed on 8 June 2022), it would be of interest to know whether patients in the study by Ceruti and colleagues were on epoprostenol treatment.

Overall, it would be critical to learn from Dr. Ceruti and colleagues how they may explain the notable differences between their published values and our calculations, and we are looking forward to their response.

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
