# Peer review of "Comment on Ceruti et al. Temporal Changes in the Oxyhemoglobin Dissociation Curve of Critically Ill COVID-19 Patients. J. Clin. Med. 2022, 11, 788"

_jcm, 2022, doi:10.3390/jcm11123376_

Round 1

Reviewer 1 Report

Böning et al ask a relevant question and do so in an eloquent letter. I have not much to add apart from very minor suggestions.

Woyke et al recently showed a left shift caused by Epoprostenol (10.1152/ajplung.00084.2022) in an ex vivo experiment. It might be an interesting additional question to ask Ceruti et al whether they used this drug. (N.B.: it was not used in the study population investigated by Vogel et al)

I t appears as if the order of references in the comment doesn’t correspond with the list of references at the end. The latter is in alphabetic order. 

Line 34 & 43: ‘for’ many years/decades (instead of ‘since’ many years/decades)

Author Response

We thank the reviewer for his suggestion. We have added a remark about epoprostenol together with a reference and a question to Dr. Ceruti about the possible use of this substance for treatment in his patients.